# Interventions for the prevention of spontaneous preterm birth: a scoping review of systematic reviews

Fiona Campbell ,[1] Shumona Salam,[2] Anthea Sutton,[1]
Shamanthi Maya Jayasooriya ,[3] Caroline Mitchell ,[3] Emmanuel Amabebe,[2]
Julie Balen,[1] Bronwen M Gillespie,[2] Kerry Parris,[2] Priya Soma-Pillay,[4]
Lawrence Chauke,[5] Brenda Narice,[2] Dilichukwu O Anumba[2]

¹ScHARR, The University of Sheffield, Sheffield, UK
²Department of Oncology and Metabolism, The University of Sheffield, Sheffield, UK
³Academic Unit of Primary Medical Care, The University of Sheffield, Sheffield, UK
⁴Steve Biko Academic Hospital, University of Pretoria, Pretoria, South Africa
⁵Department of Obstetrics and Gynaecology, University of Witwatersrand, Johannesburg, South Africa

**Correspondence to**
Dr Fiona Campbell;
f.campbell@sheffield.ac.uk

## ABSTRACT

**Background** Globally, 11% of babies are born preterm each year. Preterm birth (PTB) is a leading cause of neonatal death and under-five mortality and morbidity, with lifelong sequelae in those who survive. PTB disproportionately impacts low/middle-income countries (LMICs) where the burden is highest.

**Objectives** This scoping review sought to the evidence for interventions that reduce the risk of PTB, focusing on the evidence from LMICs and describing how context is considered in evidence synthesis.

**Design** We conducted a scoping review, to describe this wide topic area. We searched five electronic databases (2009–2020) and contacted experts to identify relevant systematic reviews of interventions to reduce the risk of PTB. We included published systematic reviews that examined the effectiveness of interventions and their effect on reducing the risk of PTB. Data were extracted and is described narratively.

**Results** 139 published systematic reviews were included in the review. Interventions were categorised as primary or secondary. The interventions where the results showed a greater effect size and consistency across review findings included treatment of syphilis and vaginal candidiasis, vitamin D supplementation and cervical cerclage. Included in the 139 reviews were 1372 unique primary source studies. 28% primary studies were undertaken in LMIC contexts and only 4.5% undertaken in a low-income country (LIC) Only 10.8% of the reviews sought to explore the impact of context on findings, and 19.4% reviews did not report the settings or the primary studies.

**Conclusion** This scoping review highlights the lack of research evidence derived from contexts where the burden of PTB globally is greatest. The lack of rigour in addressing contextual applicability within systematic review methods is also highlighted. This presents a risk of inappropriate and unsafe recommendations for practice within these contexts. It also highlights a need for primary research, developing and testing interventions in LIC settings.

## BACKGROUND

Preterm birth (PTB) is a global and public health priority. It is defined by the WHO as delivery before 37 completed weeks of gestation, with extremely preterm delivery defined

### Strengths and limitations of this study

⇒ Scoping review methodology enabled us to look at a broad topic area and analyse how context is taken into account in the included systematic reviews. Primary studies not reported in systematic reviews will therefore have not been included in our analysis.

⇒ We were not able to identify the setting of all primary studies where this was not reported and there is a risk that some studies, which have multiple publications may have been double counted.

⇒ We only included systematic reviews published in English.

as occurring at less than 28 weeks, very preterm delivery occurring between 28 and 32 weeks, and moderate to late preterm delivery occurring from 32 through 36 weeks.[1] It is one of the leading causes of neonatal death and under-five mortality and morbidity, with lifelong sequelae.[2] Children born prematurely have increased risks of cognitive problems, such as academic underachievement, behavioural problems and cerebral palsy than those born at full term.[3] They are more likely to experience hospital admission due to infection, particularly during infancy.[4] For parents, the financial, social and emotional effects are devastating.[3]

The global burden of preterm birth (PTB) is falling more heavily on countries with fewer resources to manage the medical, social and economic complexities of caring for premature infants. Globally, there are approximately 15 million live PTBs each year, which is estimated to be about 11% of all deliveries each year, ranging from about 8.7% in northern Europe to 13.4% in North Africa.[5 6] The majority of PTBs occur in low/middle-income countries (LMICs).[6] The highest PTB rates in 2014 occurred in Southeast Asia, South Asia and sub-Saharan Africa.

Nine of the 11 countries with the highest rates were in Africa. Furthermore, 60% of all PTBs were estimated to have occurred in sub-Saharan Africa and South Asia accounting for just over 9 million of the almost 15 million PTBs that occurred worldwide in 2010 resulting in a PTB rate of 12.8% in those settings.

Patterns of PTB differ between high-income countries (HICs) and LMICs. However, the differences in these patterns, causes and distribution of PTB is unclear and have not been fully explored. PTB is multifactorial in its aetiology and has distinct biological pathways. The aetiologies differ according to gestational age, ethnicity and characteristics unique to each population. In order to redress the burden of PTB in LMICs, additional insight into the causative and associated factors in these settings is required.

While a number of reviews and overviews of reviews of interventions to reduce the risk of PTB have been undertaken,[7–10] none have explored how many of the primary studies included in these reviews were undertaken in LMIC contexts. It is clear that some interventions that are effective in HIC contexts but may be harmful in LMIC settings, such as the use of antenatal corticosteroids[11] and cerclage.[12] It is also possible that treatments effective in HIC contexts may be even more beneficial or appropriate in LMIC contexts, such as nutritional supplements, interventions to increase birth spacing or interventions to improve the accuracy of measuring gestational age.

We have undertaken a broad scoping review of systematic reviews on interventions to reduce the risk of PTB identifying primary studies undertaken in LMICs. This will allow us to identify potential areas for further synthesis of the evidence and also to identify gaps in the research in order to direct future primary research.

### Review objectives

1. To identify systematic reviews that have sought to explore the effectiveness, safety and acceptability of interventions to prevent PTB.
2. To map research evidence to global settings to identify the geographical and economic contexts in which evidence is derived.
3. To identify where gaps in the research base exist (for real world, effectiveness, pragmatic studies) in LMIC contexts to inform future research and to generate research priorities.
4. To describe the methods used in meta-analysis to take into account geographical and regional differences in PTB.

### METHODS

We used a scoping review methodology[13] to describe the existing evidence (systematic reviews) available across primary and secondary interventions to prevent PTB, published between 2009 and 2020. Systematic scoping draws on methods described by Arksey and O'Malley[14] for scoping reviews: '[…] a form of knowledge synthesis that addresses an exploratory research question aimed at scoping key concepts, types of evidence, and gaps in research related to a defined area or field by systematically searching, selecting, and synthesizing existing knowledge'.[14] The approach enabled us to highlight the evidence gap and to assist with simultaneously undertaking a research prioritisation exercise and guideline development, as well as to inform a broader programme of research that aimed to develop effective postnatal interventions to mitigate PTB in LMIC settings. It also enabled us to generate a mega-map, an interactive table supported on our project website and designed as a visual tool to identify research gaps and facilitate ready access to relevant evidence (https://www.primeglobalhealth.co.uk/evidence-map-2-7-2020.html).

### Identifying relevant studies

Relevant systematic reviews were identified by systematic searches in the following electronic databases: Ovid MEDLINE, Cochrane Database of Systematic Reviews, PsycINFO via Ovid, EMBASE via Ovid and CINAHL via EBSCO. Each database was searched using the database thesaurus and the key word/free text method with terms relating to PTB combined with a systematic reviews filter. The search strategy incorporated the following limitations: articles written in English, and Human studies only from April 2009 to July 2020. Relevant systematic reviews were identified by systematic searches in the following electronic databases: MEDLINE, The Cochrane Library, PsycINFO, EMBASE and CINAHL. Each database was searched using the database thesaurus and the key word/free text method. The search strategy incorporated the following limitations: articles written in English, and Human studies only from April 2009 to July 2020. The date limit was selected due to the existence of a previous review for which the studies were conducted in April 2009.[15] Full search strategies have been described and published.[16]

We began with a framework of interventions identified by two existing reviews[7 8] as these were broad in their focus and encompassed a range of interventions. Any new intervention types identified during the screening process were then added to the map.

The process of study selection was based on inclusion and exclusion criteria as described in box 1. After removal of duplicates and irrelevant studies, based on the titles and abstracts, all potentially relevant reviews were read in full. Citations were screened by two reviewers (FC and one of the following team members SS, SMJ, EA, JB, BMG, BN, KP) independently and differences were resolved by discussion.

### Data extraction and coding

Data were extracted using an agreed and piloted template and coded in Excel by two reviewers working independently (FC and one of the following team members SS, SMJ, EA, JB, BMG, BN, KP) differences were resolved by discussion. The following data categories were extracted:

## Box 1 Inclusion/exclusion criteria based on PICOS

**Population**
⇒ Pregnant women at less than 37 completed weeks gestation without signs of threatened preterm labour or premature rupture of membranes.
⇒ Excluded reviews where the study population was defined by comorbidities.

**Intervention**
⇒ All interventions deliverable during pregnancy to prevent spontaneous preterm birth (PTB) (these included clinical, behavioural and nutritional interventions and health systems and policy interventions).
⇒ All interventions assessed the risk of PTB.
⇒ Excluded interventions given to pregnant women to improve neonatal outcomes.

**Comparators**
⇒ We included any comparator, including placebo or alternative treatments.

**Outcomes**
⇒ We included reviews which focused on PTB as an outcome.
⇒ Where it is reported, we state how many of the primary studies measured PTB as an outcome and the resulting data used in the synthesis.

**Study design**
⇒ Systematic reviews published between April 2009 and July 2020, of studies that have evaluated interventions to prevent PTB, or that measured PTB as a relevant outcome.

**Outcomes**
⇒ PTB (<28, <34, <37 weeks gestation) .
⇒ We recorded neonatal outcomes and adverse outcomes if reported within the review.

number of included studies, review PICO, setting of primary studies and any analysis that took into account study setting or population characteristics, PTB outcomes, assessment of adverse effects and recommendations for practice and research. PTB rates in low-income countries (LICs), lower middle-income countries (LMCs), upper middle-income countries (UMCs) and HICs settings were drawn from data published in a rigorous review of national civil registration and vital statistics to determine global, regional and national estimates of levels of PTB.[6]

Where reported information allowed, we used the World Bank categories to identify the categories of all country settings identified in the reviews.[17]

The population, interventions, comparators, outcomes and reviewer conclusions for future research were tabulated and described narratively. The country or countries of the included primary studies were noted, and the methods used in the review for analyses of data from different settings was also recorded and described. We did not contact review authors for missing data.

### Patient and public involvement

This review was undertaken as part of a larger programme of research in PTB (NIHR Global Health under grant (17/63/26)). The programme iPatient and public involvements informed by key stakeholders and a patient and public involvement (PPI) advisory group comprising representatives from Sheffield, Bangladesh, and South Africa. The design and questions for the review were informed by consultation with these groups.

## RESULTS

Our search identified 3133 citations which were screened by two reviewers. A third reviewer was also involved where there was a lack of consensus or uncertainty regarding inclusion. Following screening, 424 full text papers were retrieved for data extraction. At data extraction a further 285 were excluded. The process of identifying the included reviews is summarised in figure 1.

We included 139 reviews which addressed a range of primary and secondary interventions and measured the effectiveness of the intervention in reducing the risk of PTB. These are summarised in table 1. There was a considerable variation in the number of included studies in the reviews for each intervention, reflecting differing research questions objectives (therefore different PICOs) and search strategies.

### Context of primary studies

A total of 1372 primary studies were included across all of the 139 reviews Not all of these studies will have been measuring PTB as an outcome but were included within the review which may have been measuring a range of maternal outcomes including PTB. The largest number of primary studies were those evaluating micronutrient supplements (n=481) and tocolytics (n=167). A total of 113 of the reviews described the country in which the primary studies were undertaken and so these data were known for 1288 (93.9%) of 1372 included primary studies. Of these, 390 (30.3%) were undertaken in LMICs, 15 primary studies were multicentre and included data gathered from LMIC and HIC settings, though only 3 of these studies included LICs. Of the studies undertaken in LMICs, a majority (n=255) examined the effects of nutritional supplements. Excluding nutritional intervention studies, the proportion of LMIC-based primary studies of interventions to reduce PTB accounts for only (n=135) 10.5% of the included studies where settings are known.

Of the total number of primary studies undertaken in LMIC contexts, those studies undertaken in LIC settings represented a very small proportion of included studies. Participants from LICs were represented in only 4.5% (n=58) of the total number of studies, and if the nutritional intervention studies are excluded, they account for only 2.5% (n=32) of the studies evaluating interventions. Of those primary studies that were undertaken in LMIC settings the numbers within each country category differed significantly. The proportion of the studies that are undertaken in LIC, LMC and UMC were 14.9% (n=58), 34.8% (n=136) and 50.2% (n=196), respectively. There are only single trials that have evaluated the impact

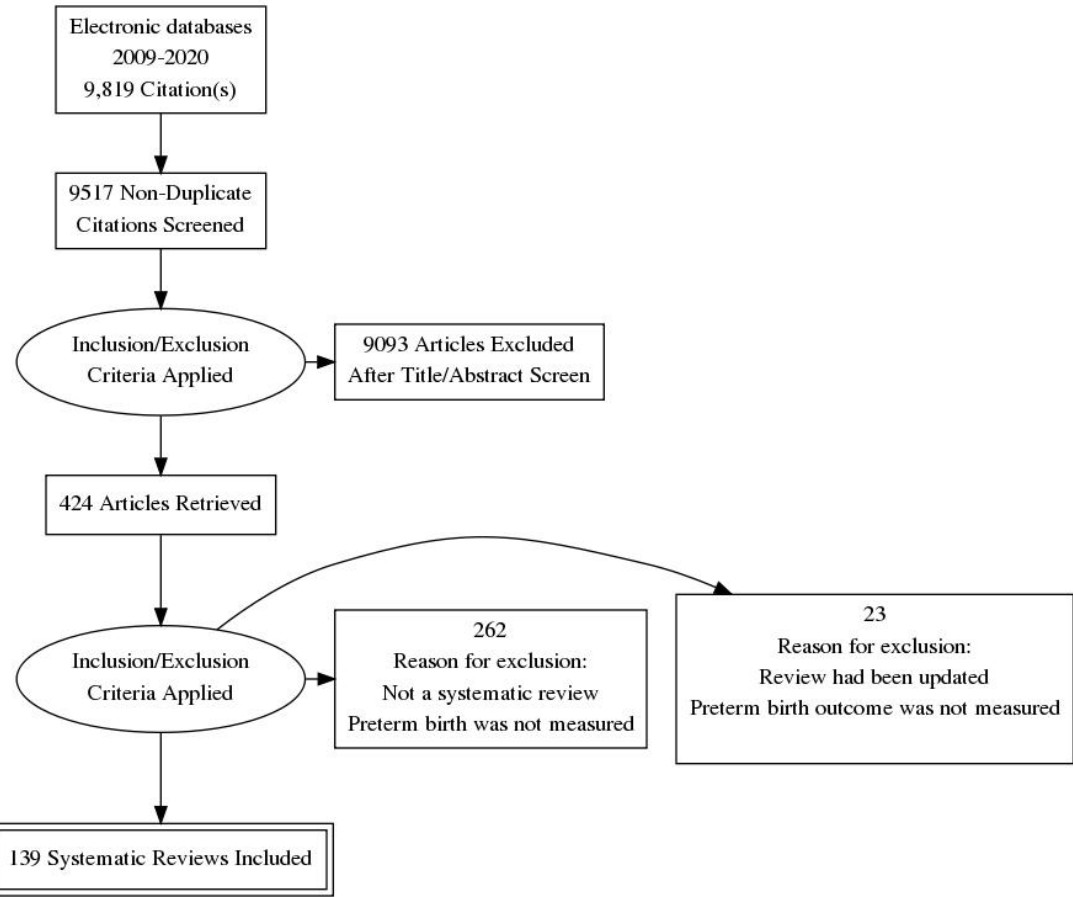

**Figure 1**  Flow of studies through review process.

of progesterone, tocolytics and interventions to increase calorie intake in LIC settings. There are no trials that have evaluated smoking cessation, preventing excessive weight gain, prevention and treatment of periodontal disease, influenza vaccine and cervical pessaries. The number of trials in each of the country categories within each intervention type are shown in table 1.

When these data are compared alongside data that shows the prevalence of PTB globally it is clear that there is an inverse pattern in the distribution of the data (figure 2).

### Effectiveness of interventions

The effectiveness of interventions in reducing the risk of PTB was variable with no intervention showing consistent effectiveness across the included reviews. Although interpretation of these data is limited by the lack of quality appraisal of the included reviews, and therefore should be viewed with caution. Overall, the scoping review demonstrates considerable inconsistency of results of interventions. Of the 139 reviews, 28 reported a reduction in PTB in intervention versus control, 80% (n=111) of the reviews found that the intervention had no impact in reducing the risk of PTB. The summary result (relative risk (RR) and OR are shown in figure 3). The results show the reduction in PTB less than 37 weeks gestation. In three reviews the intervention was not statistically

significant at 37 weeks but was reported as statistically significant at 34 weeks,[18] 35 weeks[19] and 36 weeks[20]. Two reviews reported a positive effect of the intervention in reducing risk of PTB but reported the outcome on a continuous measure. These included the effectiveness of macronutrient supplements[21] (SMD −0.19 (95% CI −0.34 to −0.04)) and cerclage (mean difference 95% CI 33.98 days (17.88 to 50.08)).[22] The interventions reporting binary outcomes which appear to have the greatest effect (RR=0.2–0.4) in reducing PTB are: antibiotics for asymptomatic bacteriuria[23] (RR=0.34 (95% CI 0.11 to 0.62), the screening and treatment of syphilis[24] (RR=0.36 (95% CI 0.27 to 0.47), and treatment of vaginal candidiasis[25] (RR=0.36, (95% CI 0.17 to 0.75). Interventions with moderate effects (RR=0.4–0.6) included treating lower genital tract infection[26] and vitamin D supplements.[27] Four of the reviews (figure 2) with a positive effect of the intervention considered that the strength of evidence supporting the finding could be considered high and the finding reliable. None of these reviews included studies conducted in LIC settings, and only one included one study in an LMIC.

### Dealing with context and generalisability within evidence synthesis

The authors of the included reviews used different approaches to dealing with the contextual variation when

**Table 1** Summary of included systematic reviews and settings of primary studies included in the review

| Interventions | Number of reviews | Number of primary studies | Country NR | Country of primary study LI | LM | UM | HI | Mixed | Studies where setting NK |
|---|---|---|---|---|---|---|---|---|---|
| **Primary prevention interventions** | | | | | | | | | |
| **Health systems** | | | | | | | | | |
| Models of antenatal care delivery (group/specialised)[61–71] | 11 | 68 | 2 | 0 | 2 | 2 | 64 | 0 | 0 |
| Midwifery led care[72] | 1 | 15 | 0 | 0 | 0 | 0 | 15 | 0 | 0 |
| Improving ANC coverage[28] | 1 | 34 | 0 | 10 | 15 | 5 | 0 | 0 | 0 |
| **Health behaviours** | | | | | | | | | |
| Smoking cessation[35 73] | 2 | 111 | 0 | 0 | 0 | 1 | 110 | 0 | 0 |
| Weight management[21 74–78] | 6 | 70 | 1 | 0 | 2 | 8 | 60 | 0 | 0 |
| **Nutritional interventions** | | | | | | | | | |
| Macronutrient supplements[29 30] | 2 | 34 | 0 | 3 | 9 | 10 | 8 | 4 | 0 |
| Micronutrient supplements[21 27–31 35–40 67–86] | 33 | 481 | 2 | 29 | 82 | 122 | 214 | 6 | 9 |
| Vitamin D[27 31 36 79–81] | 6 | 75 | | | | | | | |
| Vitamin A[37 82] | 2 | 24 | | | | | | | |
| Vitamin E, C, E and C[38 39 83] | 3 | 67 | | | | | | | |
| Iron, folic acid, iron and folic acid[40 84–90] | 8 | 182 | | | | | | | |
| Fish oil[91–95] | 5 | 38 | | | | | | | |
| Zinc[32 96] | 2 | 25 | | | | | | | |
| Calcium[41 42] | 2 | 27 | | | | | | | |
| Iodine[97] | 2 | 14 | | | | | | | |
| Multiple micronutrients[43 44 98] | 3 | 29 | | | | | | | |
| **Screening and treatment of periodontal disease[99–110]** | 12 | 46 | 0 | 0 | 3 | 7 | 36 | 0 | 0 |
| **Screening and prevention/treatment of infection** | 14 | 91 | 2 | 2 | 2 | 6 | 79 | 0 | 2 |
| Asymptomatic bacteriuria[23 111–113] | 4 | | | | | | | | |
| Screening and antibiotics for syphilis[24] | 1 | | | | | | | | |
| Influenza vaccine[114 115] | 2 | | | | | | | | |
| Lower genital tract infection[26] | 1 | | | | | | | | |
| UTI[116 117] | 2 | | | | | | | | |
| Vaginal candidiasis[25] | 1 | | | | | | | | |
| Non-specific infection[118 119] | 2 | | | | | | | | |
| Malaria[33 120 121] | 3 | 17 | 0 | 8 | 7 | 2 | 2 | 0 | 0 |
| **Secondary prevention interventions** | | | | | | | | | |
| Cerclage[18 22 45 122–136] | 18 | 123 | 10 | 0 | 7 | 11 | 42 | | 51 |
| Bed rest[137–139] | 3 | 40 | 1 | 4 | 0 | 0 | 36 | 0 | 0 |
| Cervical pessary[140–145] | 6 | 16 | 0 | 0 | 0 | 1 | 14 | 1 | 0 |
| Progesterone[19 20 146–159] | 16 | 59 | 5 | 1 | 7 | 8 | 28 | 4 | 11 |
| Tocolytics[160–172] | 11 | 167 | 3 | 1 | 0 | 13 | 68 | 0 | 84 |

ANC, antenatal care; HI, high income; LI, low income; LM, low middle; NK, not known; NR, not reported; UM, upper middle; UTI, urinary tract infection.

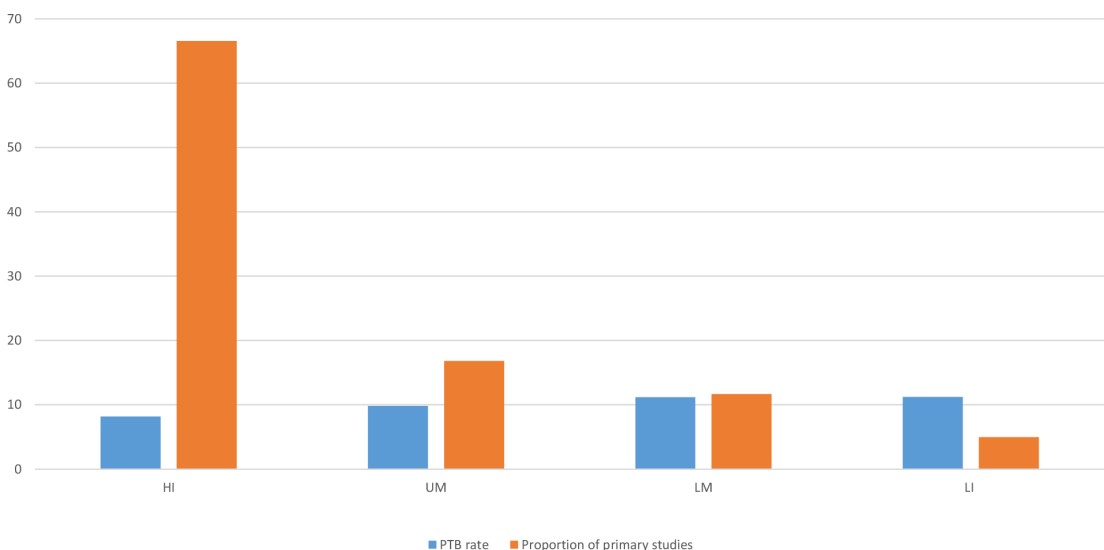

**Figure 2** Rates of PTB and proportion of primary studies undertaken in each setting. HI, high income; LI, low income; LM, low middle; PTB, preterm birth; UM, upper middle.

pooling data from primary studies, which was either to ignore, document, explore or control for differences. Twenty-seven reviews (23.8%) did not describe the setting of the primary study, ignoring variation in outcomes that may arise as a result of these differences. This occurred most frequently in reviews of cervical cerclage (see table 1). The majority of the included reviews 86 (76.1%) documented the country in which the primary study was carried out either within the text, tables of study characteristics or in accompanying appendices, but this was not considered further in terms of its implications for the findings, or application for future practice or research.

Eight reviews[27–34] sought to explore the impact of geographical and economic context by undertaking a subgroup analysis comparing trials conducted in low income settings with those in high income settings or regression analysis with geographical regions as covariates (Africa, Americas, Southeast Asia, Europe, Eastern Mediterranean, Western Pacific). In addition, one study[34] listed the country instead of the author name on the forest plot allowing ready visualisation of differences across settings. Nine reviews[35–43] undertook subgroup analysis based on features of the population that might vary across settings and influence the effectiveness of the

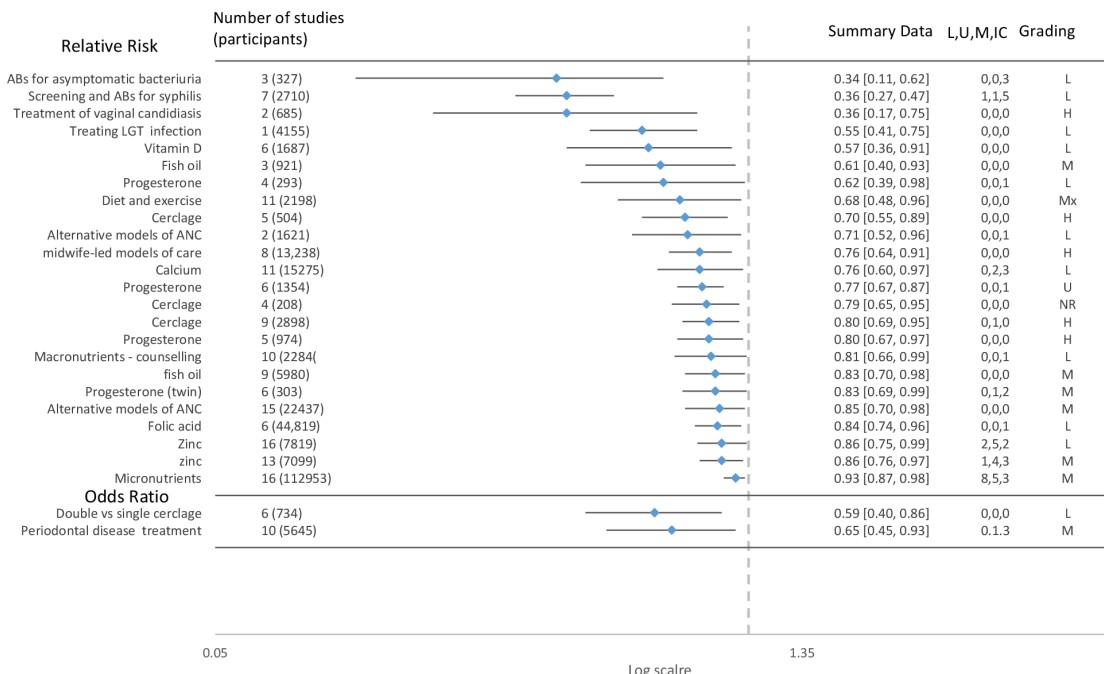

**Figure 3** Summary results of systematic reviews of interventions showing reduction in risk of preterm birth. ANC, antenatal care; L, U, M, IC, low, low middle, upper middle-income countries; LGT, lower genital tract; RR, relative risk.

intervention, such as baseline nutritional status of the mother. One review[44] exploring multiple micronutrient supplementation controlled for settings by limiting the review to include only those studies undertaken in LMIC contexts. Four reviews[19 20 44 45] undertook an IPD (individual patient data) analysis, allowing subgroup analyses about differences in effect more easily than with aggregate data. This approach allowed comparison between effects for women recruited and receiving the intervention in different settings, effect sizes in each country could also be shown in the analyses.

## DISCUSSION

This scoping review has revealed an inverse pattern of research, with only 30.3% of published research included in systematic reviews of interventions reporting PTB outcomes carried out in LMIC settings, and only 4.5% was conducted in the poorest countries in the world where the burden of PTB is greatest. The distribution of types of intervention tested and evaluated in these settings is not even across interventions, but is largely focused on very context specific interventions (prevention of malarial infection) and nutritional supplementation. Similar patterns of a mismatch between research effort and health needs in non-high income regions have been identified across a broad range of diseases.[46 47] It has also been previously reported that primary research often fails to capture those with the greatest healthcare needs such as vulnerable populations.[48 49]

This review has also revealed a limited approach in evidence synthesis to explore the applicability of findings across geographical settings and to draw attention to these gaps with a resultant risk that interventions shown to be effective in HI settings may not translate to LIC settings and may indeed have adverse effects when applied to LIC settings. Likewise, the focus of research in HIC settings means that interventions that may have greater benefit in LIC settings—where the problem is greatest—remain untested or replicated with larger numbers of participants. Adolescent pregnancy and short inter pregnancy intervals, both of which are more common in LMICs, have been highlighted as important risk factors for PTB[50] yet there is a lack of data on interventions to address these and their effectiveness in reducing the risk of PTB.

The lack of robust evidence to inform both the primary and secondary prevention of PTB in LIC settings, where the prevalence of PTB is highest presents challenges for developing appropriate and contextually relevant clinical guidance. The factors that mean findings cannot be generalised from high resource settings to low and middle resource settings are multiple and will differ across interventions. Ethnicity, poverty, gender dynamics, pollution, temperature, climate, diet, access to healthcare, educational status, employment conditions are all examples of factors that might play a role in these differences. Improved understanding of the aetiopathogenesis of PTB is also necessary for defining an accurate model of risk prediction and would help in understanding what factors in local settings increase risk and facilitate the development of an accurate model of risk prediction.[51]

Two recent overviews of reviews[9 10] also found that few interventions are effective in PTB prevention. The following interventions were identified in these reviews as showing positive or possible benefit: lifestyle and behavioural changes (including diet and exercise); nutritional supplements (including calcium, zinc and vitamin D supplementation); nutritional education; and screening for lower genital tract infections. Positive effects of secondary interventions were found for low dose aspirin among women at risk of pre-eclampsia; clindamycin for treatment of bacterial vaginosis; treatment of vaginal candidiasis; progesterone in women with prior spontaneous PTB and in those with short mid-trimester cervical length; L-arginine in women at risk for pre-eclampsia; levothyroxine among women with thyroid disease; calcium supplementation in women at risk of hypertensive disorders; smoking cessation; cervical length screening in women with history of PTB with placement of cerclage in those with short cervix; cervical pessary in singleton gestations with short cervix; and treatment of periodontal disease. Our review findings were in concordance, although, in addition, we identified screening and antibiotic treatment for syphilis, and positive effects of fish oil supplements. In most instances the trials were small and authors recommended larger well-designed randomised controlled trials (RCTs). The lack of consistency across review findings for interventions also merits more exploration. Compromised methodological rigour can inflate trial findings by 30%–50%.[52 53] Some of the differences in our review findings reflect some differences in the included reviews.

The interventions identified in this review, and those of Matei *et al*[9] and Medley *et al*[10] informing guideline development, clinical practice and policy decision making have been little tested in LMIC settings. In those interventions where there is more consistency in review findings such as cervical cerclage, there are no studies that have been conducted in low-income settings and over half of the reviews did not report or consider settings in their analyses.

This scoping review has shown that many authors of systematic reviews fail to use design and statistical approaches that adequately address contextual variations between the included source studies and imperfectly represent 'real world' conditions within the target context. While those reviews that sought to take into account LMIC contexts were unable to conduct the analyses due to a lack of data, they nonetheless were able to highlight the gaps in research, for example the lack of studies in vitamin D undertaken in Africa.[31]

The Preferred Reporting Items for Systematic reviews and Meta-Analyses (PRISMA) reporting standards reference 'context' in terms of the circumstances requiring the review itself, rather than referencing the contexts of studies included in the review.[54] The PRISMA extension

for Complex Interventions includes the elements of 'time' and 'setting'.[55] However, grouping LMIC data, or even LI data may still be too broad. Even within the categories of LIC there is considerable diversity that may impact on how an intervention works and within countries there may also be considerable diversity between the wealthiest and poorest groups. For example, the time taken to reach comprehensive emergency obstetric care facilities in low resource settings is often underestimated and for most women is likely to be 120 min of travel time.[56] Context cannot be standardised, it will vary from review to review, as different interventions and different populations are considered. 'Context' and the factors that might influence the efficacy, uptake, acceptability, appropriateness, accessibility and availability of an intervention requires a good understanding of the aetiology and mechanisms by which risk factors interact with environmental, microbial, socio-political and health system variations across settings.[57]

It must be acknowledged that there are significant barriers to undertaking research in many settings across the globe. These include very practical challenges such as a lack of access to high-quality data and the challenges of estimating gestational age.[58] Recent changes to global health funding arena include a very large proportion being spent on the pandemic as well as government reductions, for example, in the UK.[59] These reductions in funding will undermine what has been a growth in research in LMIC settings and will impede efforts to address the imbalances highlighted in this scoping review.

A number of limitations exist in this scoping review. We have not sought to identify the setting of primary studies where this is not reported in the systematic review. We have also not limited our analysis to studies within the reviews that only contributed findings to the risk of PTB. Most reviews explored several maternal and infant outcomes. Therefore, in this scoping review, included primary studies may not have contained PTB outcome data. We limited our scoping review to exploring evidence within systematic reviews as these are key sources of evidence to inform guideline development and policy decision making. It is possible that further primary studies have been published but are not included in this analysis. Nevertheless, it gives an indication of the distribution of research being undertaken in the poorest regions of the world that address PTB.

## CONCLUSION

Only 4.5% of primary research to examine the effectiveness of interventions to reduce the risk of PTB is carried out in settings where the burden is greatest. No interventions which reduce the risk of PTB, judged to be supported by strong evidence, include studies undertaken in low resource settings. In the synthesis of studies, current methods often fail to address the

contextual variation and consider the applicability of findings in low resource, high burden settings. This has implications for supporting policy making, and development of contextually relevant clinical guidelines. While methods can be undertaken to improve approaches to evidence synthesis, they cannot compensate for the lack of primary research in low resource settings. This is critical if global health inequalities are to be addressed and millennium development goals[60] to reduce under-five mortality are to be achieved. Funding and supporting research in LMICs would have a threefold benefit; first, if the prevalence of the disease is higher it is easier to reach statistical significance for efficacy or inefficacy of each tested intervention. Second, it would address the knowledge gap highlighted in this review and finally—and most importantly—the implementation of effective interventions would have the potential for greater public health impact where the risks are greater, more prevalent and outcomes more severe.

**Contributors** FC, PS-P, LC prepared the protocol, AS designed search strategies, SS, SMJ, CM, EA, JB, BMG, KP, BN collected and analysed or interpreted data. FC prepared manuscript. SS, AS, SMJ, CM, EA, JB, BMG, KP, PS-P, LC, BN, DOA edited or read and approved the final manuscript. FC is the guarantor and accepts full responsibility for the conduct of this study.

**Funding** NIHR provided funding for researcher support.

**Competing interests** None declared.

**Patient and public involvement** Patients and/or the public were involved in the design, or conduct, or reporting, or dissemination plans of this research. Refer to the Methods section for further details.

**Patient consent for publication** Not applicable.

**Ethics approval** Not applicable.

**Provenance and peer review** Not commissioned; externally peer reviewed.

**Data availability statement** Data are available upon reasonable request. All data extracted from the included reviews is available on request from the corresponding author.

**ORCID iDs**
Fiona Campbell http://orcid.org/0000-0002-4141-8863
Shamanthi Maya Jayasooriya http://orcid.org/0000-0002-1147-5744
Caroline Mitchell http://orcid.org/0000-0002-4790-0095

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
