## [Reviewer comments · BMJ Open]

ARTICLE DETAILS

TITLE (PROVISIONAL)	Interventions for the prevention of spontaneous preterm birth: a scoping review of systematic reviews
AUTHORS	Campbell, Fiona; Salam, Shumona; Sutton, Anthea; Jayasooriya, Shamanthi; Mitchell, Caroline; Amabebe, Emmanuel; Balen, Julie; Gillespie, Bronwen; Parris, Kerry; Soma-Pillay, Priya; Chauke, Lawrence; Narice, Brenda; Anumba, Dilichukwu

VERSION 1 – REVIEW

REVIEWER	Cavoretto, Paolo Univ Vita Salute San Raffaele, Obstetrics and Gynaecology
REVIEW RETURNED	17-Oct-2021

GENERAL COMMENTS	This is a scoping review study assessing Spontaneous Preterm Birth Prevention Highlighting an Inverse Pattern of Research with a Lack of Research Evidence from High Burden Settings, such as low and middle-income countries. The authors collected 139 systematic review on preventive interventions for PTB and they highlighted the lack of research evidence derived from contexts where the burden of PTB globally is greatest (low and middle income countries). The authors made a call for performing primary study in such settings. The study is interesting, scientifically sound and well-written. I have few constructive criticisms for the authors before publication: 1. Please consider discussing new methodologies for selecting risk factors for PTB based upon multilevel statistical methods including machine learning and citing appropriate references (1). This may well apply to the interest of the authors, since these methods may overcome the local setting in which the study is carried out contributing to increase generalizability of results. This would be the basis for future interventions on those risk factors. It does not diminish the need for primary studies as stated by the authors.2. I recommend to the authors to raise a call for international bodies funding research grants to promote studies to be performed in the low and middle income countries. This would have a three-fold benefit for research: firstly, if the prevalence of the disease is higher it is easier to reach statistical significance for efficacy or inefficacy of each tested intervention (higher statistical power); secondly, these researches would fill the gap of knowledge found by the authors; finally and not lastly the development of research would contribute to improve assistance and potentially outcome of low resources settings with a great potential benefits for individual patients, as well as the whole population.3. I noted that the authors did not choose to follow the scheme of PRISMA for scoping review in their article structures as far as the headers and chapters nomenclature (2)
---

	4. The authors must discuss more why they believe that conclusions achieved within studies in high income settings cannot be generalized and extended to low and middle resources settings. I believe this is neither just a matter of ethnicity (which may be addressed with subgroup analyses of studies performed in high resources settings on specific ethnic minorities), nor a matter of gross money income or educational status, but more a mixture of all of these. In addition, specific other covariates may also be adding relevance to the authors concern (which I fully share and approve) such as diet, working habits, environmental issue (pollution, etc), temperature and climate. This issue must find some room in the manuscript to justify the authors concern on generalizability of available studies. 5. Study limitations are not presented by the authors. A major limitation of this study is the lack of proposals and suggestions to overcome the described paucity of evidences arising from the low and middle income countries. References: 1. Della Rosa PA, et a. A hierarchical procedure to select intrauterine and extrauterine factors for methodological validation of preterm birth risk estimation. BMC Pregnancy Childbirth. 2021 Apr 16;21(1):306. doi: 10.1186/s12884-021-03654-3. PMID: 33863296; PMCID: PMC8052693. 2. http://prisma-statement.org/documents/PRISMA-ScR-Fillable-Checklist_11Sept2019.pdf
--	---

REVIEWER	Rahman, M University of Malaysia Sarawak, Community Medicine and Public Health
REVIEW RETURNED	10-Dec-2021

GENERAL COMMENTS	Overall, the scoping review article is good, but the authors tried to grasp too many objectives. The presentation of results is not according to objectives. Need major correction and rewrite the whole. Follow my comments and suggestions in the text.
---

VERSION 1 – AUTHOR RESPONSE

Reviewer 1

1)	This is a scoping review study assessing Spontaneous Preterm Birth Prevention Highlighting an Inverse Pattern of Research with a Lack of Research Evidence from High Burden Settings, such as low and middle-income countries. The authors collected 139 systematic review on preventive interventions for PTB and they highlighted the lack of research evidence derived from contexts where the burden of PTB globally is greatest (low and middle income countries). The authors made a call for performing primary study in such settings.	
2)	The study is interesting, scientifically sound and well-written. I have few constructive criticisms for the authors before publication:	Thank you
3)	1. Please consider discussing new methodologies for selecting risk factors for PTB based upon multilevel	Thank you for this helpful comment and

	statistical methods including machine learning and citing appropriate references (1). This may well apply to the interest of the authors, since these methods may overcome the local setting in which the study is carried out contributing to increase generalizability of results. This would be the basis for future interventions on those risk factors. It does not diminish the need for primary studies as stated by the authors.	reference. These have been added to the text.
4)	2. I recommend to the authors to raise a call for international bodies funding research grants to promote studies to be performed in the low and middle income countries. This would have a three-fold benefit for research: firstly, if the prevalence of the disease is higher it is easier to reach statistical significance for efficacy or inefficacy of each tested intervention (higher statistical power); secondly, these researches would fill the gap of knowledge found by the authors; finally and not lastly the development of research would contribute to improve assistance and potentially outcome of low resources settings with a great potential benefits for individual patients, as well as the whole population.	Thank you – we have added this as a final sentence to the manuscript
5)	3. I noted that the authors did not choose to follow the scheme of PRISMA for scoping review in their article structures as far as the headers and chapters nomenclature (2)	Thank you for this comment and reference to the PRISMA-ScR. We have used the checklist to ensure we have maintained the standards required in our reporting but have followed the journal guidance for headings. We have used the checklist to ensure we have adhered to PRISMA standards in our reporting.
6)	4. The authors must discuss more why they believe that conclusions achieved within studies in high income settings cannot be generalized and extended to low and middle resources settings. I believe this is neither just a matter of ethnicity (which may be addressed with subgroup analyses of studies performed in high resources settings on specific ethnic minorities), nor a matter of gross money income or educational status, but more a mixture of all of these. In addition, specific other covariates may also be adding relevance to the authors concern (which I fully share and approve) such as diet, working habits, environmental issue (pollution, etc), temperature and climate. This issue must find some room in the manuscript to justify the authors concern on generalizability of available studies. 1. Della Rosa PA, et a. A hierarchical procedure to select intrauterine and extrauterine factors for methodological validation of preterm birth risk estimation. BMC Pregnancy Childbirth. 2021 Apr 16;21(1):306. doi: 10.1186/s12884-021-03654-3. PMID: 33863296; PMCID: PMC8052693.	Thank you – we have added this to the text.
7)	5. Study limitations are not presented by the authors. A major limitation of this study is the lack of proposals and	Study limitations are described on page 2. The

	suggestions to overcome the described paucity of evidences arising from the low and middle income countries.	objective of the work was to describe the existing evidence, it was not within our scope to see to overcome the lack of evidence, simply to quantify it. This has not previously been done.
8)	Reviewer: 2	
9)	Overall, the scoping review article is good, but the authors tried to grasp too many objectives. The presentation of results is not according to objectives. Need major correction and rewrite the whole. Follow my comments and suggestions in the text.	Thank you for your comment but we respectfully disagree. Our objectives were (in summary)  1) To identify systematic reviews relating to spontaneous pre-term birth 2) To describe the global settings of the primary studies cited in the systematic reviews. 3) To identify gaps in evidence 4) To critique the methods used in the meta-analyses in the included reviews The results are set out as follows: Describing the identified studies and the global settings (context) from which the primary studies were undertaken (objectives 1 and 2) Describing where there was missing evidence of intervention effectiveness (objective 3) Dealing with context and generalisability within evidence synthesis (objective 4)
10)	Split first sentence	Amendment made
11)	Results in abstract - rewrite	Completed
12)	address the objectives	Please see our comment above
13)	pg 7 ref 151 check (sequence is not right)	Thank you – this has been corrected.
14)	table 1 (is it table 2)	Thank you - this has been corrected.

15)	rewrite results aligned to objectives	Please see our earlier comment
16)	Describe how MA performed.	We did not undertake a meta-analysis, Figure 3 shows the results of the meta-analyses that were reported in the included systematic reviews. The table was generated using Excel.

VERSION 2 – REVIEW

REVIEWER	Cavoretto, Paolo Univ Vita Salute San Raffaele, Obstetrics and Gynaecology
REVIEW RETURNED	06-Feb-2022

GENERAL COMMENTS	The authors successfully resolved all issues raised in my first review. The article is now acceptable for publication.
--

REVIEWER	Rahman, M University of Malaysia Sarawak, Community Medicine and Public Health
REVIEW RETURNED	19-Feb-2022

GENERAL COMMENTS	We may accept the paper for publication.
--